# Substance GP-2250 as a New Therapeutic Agent for Malignant Peritoneal Mesothelioma—A 3-D In Vitro Study

**DOI:** 10.3390/ijms23137293

**Published:** 2022-06-30

**Authors:** Claudia Baron, Marie Buchholz, Britta Majchrzak-Stiller, Ilka Peters, Daniel Fein, Thomas Müller, Waldemar Uhl, Philipp Höhn, Johanna Strotmann, Chris Braumann

**Affiliations:** 1Department of General-, Visceral-, Thoracic- and Vascular Surgery, Prosper Hospital, Stiftungsklinikum PROSELIS, 45659 Recklinghausen, Germany; claudia.baron@rub.de; 2Department of General and Visceral Surgery, St. Josef-Hospital, Ruhr-University Bochum, 44791 Bochum, Germany; marie.buchholz-a7y@rub.de (M.B.); britta.majchrzak@rub.de (B.M.-S.); ilka.peters@rub.de (I.P.); daniel.fein@kklbo.de (D.F.); waldemar.uhl@klinikum-bochum.de (W.U.); philipp.hoehn@rub.de (P.H.); chris.braumann@kklbo.de (C.B.); 3Geistlich Pharma AG, CH-6110 Wolhusen, Switzerland; thomas.mueller@geistlich.com; 4Department of General, Visceral and Vascular Surgery, Evangelische Kliniken Gelsenkirchen, Akademisches Lehrkrankenhaus der Universität Duisburg-Essen, 45878 Gelsenkirchen, Germany

**Keywords:** malignant peritoneal mesothelioma, spheroid model, in vitro, GP-2250, treatment

## Abstract

Malignant peritoneal mesothelioma is a rare tumor entity. Although cytoreductive surgery and hyperthermic intraperitoneal chemotherapy have increased overall survival, its prognosis remains poor. Established chemotherapeutics include mitomycin C (MMC) and cisplatin (CP), both characterized by severe side effects. GP-2250 is a novel antineoplastic agent, currently under clinical investigation. This in vitro study aims to investigate effects of GP-2250 including combinations with CP and MMC on malignant mesothelioma. JL-1 and MSTO-211H mesothelioma cell lines were treated with increasing doses of GP-2250, CP, MMC and combination therapies of GP-2250 + CP/MMC. Microscopic effects were documented, and a flow-cytometric apoptosis/necrosis assay was performed. Synergistic and antagonistic effects were analyzed by computing the combination index by Chou-Talalay. GP-2250 showed an antiadhesive effect on JL-1 and MSTO-211H spheroids. It had a dose-dependent cytotoxic effect on both monolayer and spheroid cultured cells, inducing apoptosis and necrosis. Combination treatments of GP-2250 with MMC and CP led to significant reductions of the effective doses of CP/MMC. Synergistic and additive effects were observed. GP-2250 showed promising antineoplastic effects on malignant mesothelioma cells in vitro especially in combination with CP/MMC. This forms the basis for further in vivo and clinical investigations in order to broaden treatment options.

## 1. Introduction

Malignant mesothelioma (MM) is a rare, aggressive tumor entity of serous membranes, the mesothelium, featuring predominantly pleura and peritoneum (peritoneal MM) [1,2]. Mesothelial cells have been described first 1827 by Bichat as monolayer flat cells in serous cavities, coating the cavities and the organs within [3]. Forming a protective layer, the mesothelium ensures free organ movement via secretion of glucosamines and lubricants but also takes part in a variety of processes including immunologic and inflammatory responses [3,4]. Possible causes of malignant degeneration include asbestos, talcum, erionite, Morbus Hodgkin, chronic peritonitis and radiation [5]. Since 1977 MM is a recognized occupational disease in Germany, and despite its low incidence, it is the most common occupational tumor entity. The average latency period between asbestos exposure and first diagnosis lies at 38.4 years [6], with latency periods ranging from 20 to 55 or even 70 years [1].

Peritoneal MM is characterized by a highly aggressive growth pattern, a poor prognosis and its resistance towards standard tumor therapy [1]. Typically, metastasis occurs per continuitatem, rarely lympho- or hematogenically. [7,8,9]. It either spreads multinodularly, with multiple distinguishable tumor nodes in the lower belly and the omentum majus, or diffusely, with plaque-like thickening of the peritoneum, walling in the abdominal organs and vessels [1]. Diagnosis of peritoneal MM and clear distinction from other abdominal tumors or peritoneal metastases is challenging. Histopathological and immunohistochemical examinations are required [1,5].

Untreated peritoneal MM leads within an average of 6 to 16 months to death after initial diagnosis. Due to its rarity, establishing a standard therapy regimen has been challenging. Currently, there are almost exclusively retrospective case reports of single institutions or multi-institutional cohort studies without a randomized controlled study design [8]. Cytoreductive surgery (CRS) with hyperthermic intraperitoneal chemotherapy (HIPEC) significantly increased the mean overall survival to 52 to 92 months [10] and has been established as standard treatment for patients with good performance status and a potentially resectable tumor [5]. There are multiple HIPEC protocols regarding application technique, duration, carrier solution and antineoplastic agents without one asserting itself over the others. Commonly used drugs include cisplatin (CP) and mitomycin C (MMC) as both agents show heat synergisms [11]. A slight benefit was documented for combination protocols involving a platin derivative [5,7,8,9,12,13]. The RENAPE study from 2018 compared 249 patients with CRS and HIPEC of different protocols. The applied substances included CP, oxaliplatin, doxorubicin, MMC and irinotecan. Although no significant difference in overall survival depending on the respective substances was found, overall survival was increased when using a combination therapy of a platin derivative with one of the other agents [14].

CRS and HIPEC utilizing established chemotherapeutics have a high rate of lesser and severe complications including pulmonal, renal and cardiac complications; hematologic toxicities and sepsis with multiorgan dysfunction, tied to the drugs’ toxicity [9]. In light of these limitations, furthering the current state of studies and searching for new options, especially concerning innovative antineoplastic agents with a low toxicity profile, is an essential task.

A recent development in antitumorigenic substances is the oxathiazinane GP-2250 (Tetrahydro-1,4,5-Oxathiazin-4,4-dioxid). Figure 1 displays the structural formula of GP-2250. The first study displaying GP-2250s anti-neoplastic potential was published 2017 by Buchholz, Majchrzak-Stiller et al. [15]. GP-2250 significantly reduced viable pancreatic carcinoma cells in vitro and in vivo in nude mice. Flow cytometry analysis showed an induction of apoptosis and necrosis. GP-2250 proved to be well tolerated, leading to acute or chronic toxicity at highly elevated concentrations of 2000 mg/kg BW and 1000 mg/kg BW, respectively, thus making it a compelling topic of study [15]. Currently the substance is subject of a clinical phase I/II study, investigating its tolerance in combination with gemcitabine in patients with advanced pancreatic carcinoma [16].

This study is the first to investigate the possible application of the novel substance GP-2250 in MM therapy.

GP-2250s cytotoxic effects on the MM cells lines JL-1 and MSTO-211H in a monolayer and spheroid model were analyzed using flow cytometry measuring the percentage portions of viable, apoptotic and necrotic. Spheroids are very dense, three-dimensional (3-D) tumor cell formations, mimicking the characteristics of solid tumors far closer than monolayer cultures and bridging the gap between in vitro monolayer cultures and in vivo studies [17,18,19,20,21,22,23]. Additionally, possible combination therapies of GP-2250 and in HIPEC protocols commonly used agents CP and MMC were investigated.

## 2. Results

### 2.1. Microscopic Effect of GP-2250 on Mesothelioma Spheroids

Both JL-1 and MSTO-211H cells formed stable spheroids (sph.), which passed through the spheroid-formations assay unaffectedly (Figure 2 and Figure 3). When incubated with GP-2250 over 48 h, both cell lines showed a dose-dependent effect of GP-2250 on the microscopic cell-clusters (Figure 2 and Figure 3). With increasing doses, the spheroid periphery appeared less dense, single cells were distinguishable within the peripheral cell cluster. Concurrently, the number of single cells on the bottom of the plate increased. A dense spheroid core was still discernible but diminished in size compared to the overall diameter with increasing dosage.

### 2.2. 48 h Treatment with GP-2250 Monotherapy Had a Cytotoxic Effect on Both MM Cell Lines

GP-2250 showed both with monolayer MM cells and with MM sph. a dose-dependent decrease in viable cells and an increase in apoptotic and necrotic cells (Figure 4 and Figure 5).

JL-1 monolayer (mnl.) cells showed a first significant reduction of viable cells after a 48 h treatment with 250 µmol/L GP-2250, MSTO-211H mnl. cells at 500 µmol/L GP-2250 (Figure 4). Maximum effect was achieved at 1000 µmol/L (JL-1 mnl.; 17.07 ± 10.06% viable cells) and 2000 µmol/L GP-2250 (MSTO-211H mnl.; 12.28 ± 1.82% viable cells). Increase in apoptotic cells considerably outweighed the rise of necrotic cells in both cell lines. Notably, in both cell lines treatment with 500 µmol/L, GP-2250 led to an indentation in the otherwise nearly linear decline of viable cells. This indentation could not be replicated with MM sph. An effective dose reducing viable cells by 50% (ED_50_) was computed at 603.7 µmol/L [321.21–1134.98 µmol/L 95% confidence interval] for JL-1 mnl. and at 1193.04 µmol/L [598.23–2379.26 µmol/L 95% confidence interval] for MSTO-211H mnl. cells (Table 1).

MM cells in spheroids showed higher resistance to GP-2250 compared to their respective monolayer cultures (Figure 5).

The first significant reduction of viable cells in JL-1 sph. was achieved by treatment with 250 µmol/L GP-2250 and with 1000 µmol/L GP-2250 in MSTO-211H sph. At a maximum dose of 1500 µmol/L GP-2250 (JL-1 sph.) and 2500 µmol/L (MSTO-211H sph.), viable cells were reduced from approximately 70% to 21.03% ± 5.78% and 43.76 ± 6.01%, respectively. An ED_50_ was computed for JL-1 sph. at 949.64 µmol/L [684.72–1317.05 µmol/L 95% confidence interval] and for MSTO-211H-sph. at 2587.99 µmol/L [1547.46–328.18 µmol/L 95% confidence interval]. In JL-1 sph., apoptotic cells outweighed necrotic cells, except for treatment with 1500 µmol/L GP-2250. In MSTO-211H sph., necrotic cells slightly outweighed apoptotic cells. The respective ED_50_’s of the GP-2250 monotherapies are shown in Table 1.

### 2.3. 48 h Combination Treatment of GP-2250 + MMC on MM Spheroids Showed Synergistic Cytotoxic Effects

Cytotoxic effects of a combination treatment of GP-2250 and MMC on both cell lines in a spheroid model were analyzed. Concentrations with low to moderate effects in monotherapy according to previous works [24] were chosen to achieve synergism using the lowest effective concentrations.

All tested combinations of GP-2250 plus MMC showed a significant decrease of viable MSTO-211H cells compared to the untreated control (Figure 6b). Apoptosis outweighed necrosis. A total of 500 and 1000 µmol/L GP-2250 were combined with either 0.5 or 2.5 µmol/L MMC, as these doses showed significant but not the strongest effects as monotherapies. The strongest reduction of viable cells was achieved using 1000 µmol/L GP-2250 + 2.5 µmol/L MMC, reducing their shares from 71.6 ± 3.7% (untreated control) to 30.5 ± 1%, equivalent to a ED_57_. Decrease under combination treatment was statistically significant compared to the respective monotherapies with 1000 µmol/L GP-2250 and 2.5 µmol/L MMC. This applied for the effect under treatment with 1000 µmol/L GP-2250 + 0.5 µmol/L MMC, as well, which equated to an ED_46_. Hence, a reduction of ED_50_ of MMC was achieved from 5–10 µmol/L MMC in monotherapy to 0.5–2.5 µmol/L MMC in a combination therapy. Both mentioned combinations additionally showed synergistic effects. Conversely, treatments with 500 µmol/L GP-2250 displayed an antagonistic (+0.5 µmol/L MMC) and additive (+2.5 µmol/L MMC) effect.

For JL-1 cells, 250 and 500 µmol/L GP-2250 were combined with either 1 or 5 µmol/L MMC (Figure 6e)). Treatment with 250 µmol/L GP-2250 + 5 µmol/L MMC and both treatments with 500 µmol/L GP-2250 reduced viable JL-1 cells extremely significantly compared to the untreated control featuring a higher percentage of apoptotic than necrotic cells. The strongest reduction was accomplished by 500 µmol/L GP-2250 + 5 µmol/L MMC from 70.2 ± 3.3% to 44.8 ± 2.9% viable cells (ED_36_). This was the only treatment displaying a significant difference in viable cells compared to its respective monotherapies. Synergistic effects were not observed, and instead, the treatments showed antagonistic effects.

Concerning the moiety of necrotic and apoptotic cells of JL-1 sph., none of the combinations proved to show a significant increase of the shares of necrotic or apoptotic cells in comparison to each monotherapy.

In MSTO-211H sph., only the combination of 1000 µmol/L GP-2250 with 0.5 µmol/L MMC led to a significant rise of apoptotic cells to 34.3% ± 6.1% in comparison to each of its monotherapies (1000 µmol/L GP-2250: 19.1% ± 7.2%, 0.5 µmol/L MMC: 21.1% ± 3.3%, *p* < 0.05, each). No such difference was observed concerning the moiety of necrotic cells.

### 2.4. 48 h Combination Treatment of GP-2250 + CP on MM-Spheroids Shows Synergistic Cytotoxic Effects

Likewise, Cytotoxic effects of a combination treatment of GP-2250 and CP on both cell lines in a spheroid model were analyzed. Concentrations with low to moderate effects in monotherapy according to previous works [25] were chosen to achieve synergism using the lowest effective concentrations.

Combination therapy of GP-2250 + CP proved to be highly effective. An ED_50_ or higher was attained for both cell lines (Table 1).

Viable MSTO-211H cells were effectively reduced with an extremely high statistical significance compared to the untreated control (Figure 6). An amount of 500 or 1000 µmol/L GP-2250 was combined with 5 or 7.5 µmol/L CP (Figure 6a). The ratio of apoptotic cells slightly surpassed necrotic cells after treatment. The highest reduction of viable cells was provoked by combinations with 1000 µmol/L GP-2250 from 71.6 ± 3.7% (untreated control) to 36.7 ± 6.3% with 5 µmol/L CP and down to 33.3 ± 2.6% with 7.5 µmol/L CP. This equated to an ED_49_ and ED_54_, respectively. Thus, the CP concentration needed to achieve ≥ED_50_ was reduced from 20 µmol/L in a monotherapy to 5–7 µmol/L CP when combined with 1000 µmol/L GP-2250. The two combinations with 1000 µmol/L GP-2250 also showed the highest statistical significance at reducing viable cells compared to their respective monotherapies and displayed synergistic effects. Viable cell reduction under treatments with 500 µmol/L GP-2250 were statistically significant as well but showed a moderate antagonistic effect.

With JL-1 sph., 250 and 500 µmol/L GP-2250 were combined with 5 and 20 µmol/L CP (Figure 6c,d). As only 500 µmol/L GP-2250 + 20 µmol/L CP caused a significant reduction of viable cells compared to untreated cultures, 30 and 40 µmol/L CP were added to the experiment. Treatment including 30 and 40 µmol/L CP proved to be extremely effective in reducing viable cells compared to the untreated control and their respective monotherapies. One exception presented the combination of 250 µmol/L GP-2250 + 40 µmol/L CP compared to a 40 µmol/L CP monotherapy, which was not statistically significant. Amounts of 250 µmol/L GP-2250 + 30 µmol/L CP and + 40 µmol/L achieved an ED_38_ and ED_40_, respectively. The strongest decrease of viable cells was caused by 500 µmol/L GP-2250 + 40 µmol/L CP from 70.2 ± 3.3% to 11.6 ± 4.9%, also displaying a moderate synergistic effect. Treatments including 30 µmol/L CP showed an additive effect. An amount of 500 µmol/L GP-2250 + 30 µmol/L CP reduced viable cells to 25.9 ± 3.9% (ED_63_). Therefore, the ≥ED_50_ of CP was lowered from 50 µmol/L in a monotherapy to 30 µmol/L in combination with 500 µmol/L GP-2250.

Concerning the moiety of apoptotic cells in JL-1 sph., only a combination of 500 µmol/L GP-2250 and 30 or 40 µmol/L, respectively, led to a significant rise of apoptotic cells (44.3 ± 3.5 and 55.9 ± 3.2%) compared to each monotherapy (30 µmol/L CP: 24.8 ± 1.6%, 40 µmol/L CP: 29.0 ± 2.7% and 500 µmol/L GP-2250: 18.7 ± 3.5%; *p* < 0.01, each), whereas no such differences were observed for the other concentration. The share of necrotic cells increased significantly for the combinations of 250 µmol/L and 500 µmol/L GP-2250 with 30 µmol/L CP (21.9 ± 0.9% and 25.0 ± 2.3%) as well as 500 µmol/L GP-2250 and 40 µmol/L CP (29.0 ± 1.6%) in comparison to each monotherapy (250 µmol GP-2250: 13.75 ± 3.5%, 500 µmol/L GP-2250: 14.9 ± 1.9%, 30 µmol/L CP: 9.9 ± 1.3%, 40 µmol/L CP: 19.7 ± 1.4%, *p* < 0.05, each).

In MSTO-211H sph., only the combination of 1000µmol GP-2250 and 5 µmol/L CP proved to significantly increase the share of apoptotic cells to 30.0 ± 2.3% versus 19.1 ± 7.2% for 1000 µmol/L GP-2250 and 19.9 ± 3.1% (*p* < 0.05, each). Concerning the moiety of necrotic cells, none of the combinations proved to significantly increase the shares in comparison to each of its monotherapies.

## 3. Discussion

This study is the first to investigate effects of the novel substance GP-2250 on MM. The substance showed dose-dependent cytotoxicity on two distinct cell lines of MM in both monolayer and spheroid models. Additionally, combination with the established chemotherapeutic agents CP and MMC proved to be highly effective in reducing cell viability and induction of apoptosis.

In previous studies, the oxathizinane GP-2250 has been shown to possess cytotoxic and proliferation-inhibiting effects on six different lines of pancreatic carcinoma cells in vitro and in vivo while being well tolerated [15]. It displayed a cell line dependent increase in effectiveness through hyperthermia up to 42 °C, rendering it most suitable for an HIPEC therapy regimen [26].

To date, no such investigation had been conducted on MM cells. This study was able to show a dose-dependent cytotoxic effect of GP-2250 on MM cells via flow cytometry for the first time.

Treatment with 250 µmol/L GP-2250 (JL-1 mnl.) and 500 µmol/L GP-2250 (MSTO-211H mnl.) led to a significant reduction of viable cells. Amounts of 1000 µmol/L GP-2250 (JL-1 mnl.) and 2000 µmol/L GP-2250 (MSTO-211H mnl.) reduced viable cells to less than 20%. This is consistent with observations of Buchholz, Majchrzak-Stiller et al. who measured the portion of vital pancreatic carcinoma cells treated with 1500 µmol/L GP-2250 at 7.6–28.5% depending on the cell line. MSTO-211H mnl. treated with the same dose had a vital percentage portion of 52.86% (±4.09%). JL-1 mnl. were not treated with 1500 µmol/L, as 1000 µmol/L GP-2250 already reduced the viable cells to 17.07% (±10.60%). In conclusion, the intensity of GP-2250s cytotoxic effect is conditional to the tumor entity and to the cell line.

After promising results using monolayer cells, further experiments were conducted using MM spheroid cultures. This 3-D model bridges the gap between monolayer in vitro studies and in vivo models as it simulates tumor microenvironment, nutrition supply, gene expression, extracellular matrix and response to treatment [18,19,20,21,22,27].

MM spheroids were treated with increasing doses of GP-2250 over 48 h. The data showed an anti-adhesive microscopic effect as well as a cytotoxic effect on MM spheroids.

Concerning cytotoxicity, a highly significant reduction of viable cells was measured beginning at 500 µmol/L (JL-1 sph.) and 1000 µmol/L GP-2250 (MSTO-211H sph.) (Figure 5). JL-1 sph. viable cells have been reduced by 50% (ED_50_) and by 70% (ED_70_) at 1000 µmol/L and 1500 µmol/L GP-2250, respectively. In contrast, MSTO-211H sph. did not reach an ED_50_ even when treated with 2500 µmol/L GP-2250. A clear change in the spheroids’ outer integrity was observed after treatment with 750 µmol/L (JL-1 sph.) and 1000 µmol/L GP-2250 (MSTO-211H sph.) (Figure 2d and Figure 3c, respectively). The lower percentage of viable cells in untreated spheroid control-cultures compared to untreated monolayer control-cultures can be explained by decreasing levels of nutrients, oxygen and cell viability in a growing 3-D cell cluster [17,23].

In the present study, the ratio of apoptotic mnl. cells outweighed the necrotic cells under treatment with GP-2250 in both, JL-1 and MSTO-211H. Similarly, JL-1 spheroids showed initially an increasing portion of apoptotic cells up to concentrations of 1500 µmol/L, where necrotic cells outweighed the apoptotic portion. At this dose, vital MSTO-211H cells were at 45.19 ± 2.02%, again with the necrotic ratio slightly higher than the apoptotic ratio. All in all, JL-1 sph. were more sensitive towards GP-2250 than MSTO-11H spheroids. To date, there are no other published studies investigating the effect of GP-2250 on tumor spheroids.

Buchholz, Majchrzak-Stiller et al. 2017 initially measured an increase of apoptotic pancreatic carcinoma cells as well, but with an increasing dose of GP-2250, they observed a preponderance of necrotic cells [15].

In monolayer, the moiety of apoptotic cells always surpassed the share of necrotic cells. In spheroids, even in untreated controls, the share of viable cells was lower, and moieties of necrotic and apoptotic cells were higher compared to monolayer cells. Additionally, the share of necrotic cells rose with increasing doses of GP-2250, surpassing the shares of apoptotic cells. This is tied to the intrinsic structure of the 3-D spheroid model where a proliferative, a dormant and a necrotic zone along a gradient of nutrients from the outside to the center can be differentiated [28].

Another explanatory approach is associated with the experimental design. Physiological apoptosis is a highly controlled, adenosine triphosphate (ATP)-dependent process with subsequent phagocytosis of the degradation products. In absence of inflammatory cells and thus phagocytosis in in vitro cell culture, secondary necrosis occurs. The presently utilized flow cytometry markers Annexin-V and Propidium Iodide (PI) cannot differentiate between primary and secondary necrosis as PI intercalates with exposed DNA, found in both necrotic and late apoptotic cells. This might lead to false positive measurement of necrotic cells.

Secondly, Buchholz, Majchrzak-Stiller et al. suggest that GP-2250 might initiate programmed necrosis. This process is a caspases-independent programmed cell death initiated by cell stress or activation of death receptors, leading to excessive levels of reactive oxygen species (ROS) [15,29]. GP-2250, in turn, has been shown to increase intracellular ROS levels [15].

Concentrations needed to achieve an ED_50_ were higher in spheroids than in monolayers of both cell lines. While in monolayer all cells are equally exposed to media and chemotherapeutics, this is not the case in 3-D structured spheroids. Thus, higher concentrations of chemotherapeutics are needed in order to expose especially the inner zones to an effective dose. Furthermore, the increased cellular adhesion of tumor cells in spheroids has been proven to reduce sensitivity towards cytotoxic drugs [18,30,31,32].

The cytotoxic effect of GP-2250 on MM spheroids was compared to the effect of standard therapeutics CP and MMC. In addition, we investigated possible combination therapies of GP-2250 plus CP or MMC.

Currently, there is no general standard therapy regimen for MM due to its small incidence and the lack of randomized multicenter studies. Most commonly used is a combination of MMC, CP or doxorubicin [11]. Although, no single regimen could assert itself over the others, regimens with a combination therapy including a platin derivate showed to be more favorable [9,14].

Both MMC and CP interact with DNA synthesis [33,34], showing strong heat synergisms [11,35]. Since their first experimental application in MM in 1984, where MMC and CP prevailed in a mice model and were successfully administered in a clinical trial [36], both agents have maintained their significance in the treatment of peritoneal MM [37]. 

CP as well as MMC are characterized by severe side effects and poor tolerability [5,13,14,38,39]. A combination therapy aiming to generate potential synergisms and thereby decrease adverse effects via dose reduction is crucial in order to ameliorate condition and quality of life in a highly vulnerable group of patients.

This is the first study to investigate a possible combination of GP-2250 with CP or MMC. When combining GP-2250 with MMC and CP, their respective doses were significantly lowered while still achieving the same decrease in viable cells. With MSTO-211H sph., ≥ED_50_ of MMC was lowered from 5–10 µmol/L to 0.5–2.5 µmol/L. Amounts of 1000 µmol/L GP-2250 plus 0.5 µmol/L MMC as well as 2.5 µmol/L MMC showed synergistic effects.

Combining a dose of 1000 µmol/L GP-2250 with CP proved most effective for MSTO-221H sph. as well. The ≥ED_50_ of CP was lowered from 10 µmol/L CP in a monotherapy to 5–7 µmol/L CP in the combination therapy with a synergistic effect. Concerning JL-1 sph., 500 µmol/L GP-2250 + 40 µmol/L CP showed a synergistic effect. An amount of 500 µmol/L GP-2250 + 30 µmol/L CP led to an additive effect and lowered the ED_50_ from 50 to 30 µmol/L CP. In total, our experiments combining GP-2250 with CP or MMC for treating MM spheroids showed promising results. Concerning synergistic effects, the present study cannot make general statements. Both combinations showed synergistic, additive as well as antagonistic effects (Table 1). 

One possible explanation for synergistic effects might be the antiadhesive effect of GP-2250 [15], lowering cell density and tissue pressure and thus leading to a deeper penetration of CP and MMC into the spheroids [40].

Another explanatory approach is an effect of GP-2250 on CD133+ cells. Those cells, on the one hand, show a heightened chemoresistance toward CP. On the other hand, cells treated with CP display an upregulation of CD133 [41]. Treatment with GP-2250 lead to a significant reduction of CD133 positive cells of the pancreatic carcinoma [26]. Lastly, CP is shown to generate oxidative stress in tumor cells through ROS. As a result, signaling pathways are initiated to trigger apoptosis, but they can also enhance chemoresistance [42]. Normally, cells regulate their ROS level intrinsically via balanced production and elimination through molecules with a thiol group. An excess of ROS can impair cellular proteins, lipids and DNA, furthering carcinogenesis but also leading to all three forms of programmed cell death [43]. MM tumor cells produce more oxidants compared to normal cells to compensate for the heightened oxidative stress [4]. ROS-driven programmed cell death is to date considered to be a main mechanism of action of GP-2250 [15]. A synergistic interference in tumor cells’ ROS homeostasis could explain synergies of GP-2250 and CP in MM spheroids.

Several limitations have to be taken into account. The present study is exclusively a cytotoxicity analysis. Direct application of in vitro data to in vivo models or even patients is not possible. Secondly the experimental design does not allow to differentiate between late apoptotic and necrotic cells. In order to fully understand and explain synergies of GP-2250 with other anti-neoplastic agents, further studies, especially in vivo, are needed.

## 4. Materials and Methods

**Cell lines and culture conditions:** Two established and commercially available human malignant mesothelioma cell lines were used in this study; biphasic MSTO-211H and epithelioid JL-1 cells. JL-1, a human epitheloid malignant mesothelioma cell line was derived from a 54-year-old male diagnosed with asbestos associated epitheloid pleural mesothelioma (DSMZ, Braunschweig, Germany). The other analysed cell line was MSTO-211H, derived from pleura effusions of a 62-year-old male diagnosed with malignant biphasic mesothelioma (ATCC, Manassas, VA, USA).

MSTO-211H cells were maintained in RPMI-1640, JL-1 cells were maintained in Dulbecco’s Modified Eagle Medium (DMEM). Both mediums were supplemented with 10% Fetal Bovine Serum (FBS), 1% Penicillin/ Streptomycin and 1% L-Glutamine (PAN Biotech GmbH, Aidenbach, Germany). Cells were maintained as monolayer in 100 mm cell culture dishes at 37 °C with 5% CO_2_ in humidified atmosphere.

**Reagents:** GP-2250 (kindly supplied by Geistlich Pharma AG, Wolhusen, Switzerland) was stored as dry powder at room temperature. It was freshly prepared every two weeks by dissolving in double distilled water, set to a physiological pH, sterile filtered and stored protected from light.

Mitomycin C (MMC) was purchased from Selleck Chemicals LLC, Munich, Germany, as dry powder and dissolved in dimethyl sulfoxide (DMSO) as described in the manufacturer specifications. The solution was divided in aliquots and stored at −20 °C. A new aliquot was used for each treatment. 

Cisplatin (CP, Hexal AG, Holzkirchen, Germany) was dissolved according to manufactures instructions in 0.9% sodium chloride and set to a physiological pH every two weeks by the St. Josef-Hospital Bochum pharmacy and was stored protected from light at 4 °C.

**Spheroid cultures and spheroid formation assay:** To generate spheroids, cells were seeded in ultra-low attachment surface 6-well plates (Corning, NY, USA) with 400.000 cells per well in 2 mL serum-free stem cell medium. The medium was composed of 95% DMEM/F12 (PAN Biotech GmbH, Aidenbach, Germany) supplemented with 2,3% Penicillin/Streptomycin (PAN Biotech GmbH, Aidenbach, Germany), 2% B27-Supplement (Thermo Fischer Scientific, Waltham, MA USA), 1% Minimal Essential Medium Non-Essential Amino Acids (Thermo Fischer Scientific, Waltham, MA USA) and 0,02% basic Fibroblast Growth Factor (Thermo Fischer Scientific, Waltham, MA USA). Spheroids were cultured for 4 days (JL-1) and 5 days (MSTO-211H) and stirred twice during that period. JL-1 spheroids were cultured for 4 days only due to their higher density. 

Afterwards, a spheroid formation assay was performed as portrayed in Figure 7. The culture was filtered over a 45 µmol/L filter to wash out singular cells. The retentate was then resuspended in 2 mL stem cell medium.

**Cytotoxicity analysis using flow cytometry:** Each treatment was conducted on 3–5 independent experiments with consecutive passages. After an incubation period of 48 h followed by microscopic documentation, a cell viability assay was performed. Monolayer cells were detached using trypsin 0.05%/ethylenediaminetetraacetic acid (EDTA) 0.02% in phosphate-buffered saline (PBS) (PAN Biotech GmbH, Aidenbach, Germany), and spheroids had to be separated using Accumax (PAN Biotech GmbH, Aidenbach, Germany). Cells were resuspended in 200 µL binding buffer (Thermo Fischer Scientific, Waltham, MA, USA). Cells were then marked with 10 µL Annexin V-FITC (Miltenyi Biotec, Bergisch Gladbach, Germany) for 15 min at room temperature protected from light. A volume of 10 µL Propidium Iodide (BD Biosciences, Franklin Lakes, NJ, USA) was added, and the percentages of viable (Annexin V-FITC and PI negative), apoptotic (Annexin V-FITC positive, PI negative) and necrotic cells (Annexin V-FITC and PI positive) were determined using flow cytometry (FACS Calibur; BD Biosciences, Heidelberg, Germany). The resulting dot plots were analyzed using CellQuest Pro software (BD Biosciences, Franklin Lakes, NJ, USA).

**Statistics:** The results of flow cytometry analysis were expressed via means ± standard deviation (SD). Comparison of means of normal distributed results of two groups was computed using the unpaired t-test by GraphPad QuickCalcs (GraphPad Software, San Diego, CA, USA). *p* values < 0.05 were considered statistically significant and marked in graphics as followed: ** p* ≤ 0.05, *** p* ≤ 0.01, **** p* ≤ 0.001. Analysis of symbiotic effects was conducted by computing the combination index (CI) by Chou-Talalay using CalcuSyn, version 2.11 (Biosoft, Cambridge, England). A CI < 0.9 was classified as a symbiotic effect (+), CI = 0.9–1.1 as an additive effect (±) and CI > 1.1 as an antagonistic effect (-). The mean effective dose was also computed by CalcuSyn. Analysis was performed as described by Chou [44,45].

## 5. Conclusions

This study proved anti-adhesive and cytotoxic effects of GP-2250 on MM monolayer cells and MM spheroids, inducing apoptosis and necrosis. Combination with CP and MMC led to synergistic and additive effects as well as to a significant reduction of ED_50_. Thus, dose reduction and thereby potentially improved tolerability and decrease of side effects at increased effects were achieved. Further studies are needed to elucidate GP-2250’s mechanism of action. Furthermore, the effect of GP-2250 on MM cells should be investigated in vivo, as well as in combination with other malignant peritoneal mesothelioma standard therapeutics such as docetaxel, irinotecan or doxorubicin. This study forms the basis for further in vivo testing and clinical trials in order to broaden treatment options in a highly vulnerable cohort of patients.

## Figures and Tables

**Figure 1 ijms-23-07293-f001:**
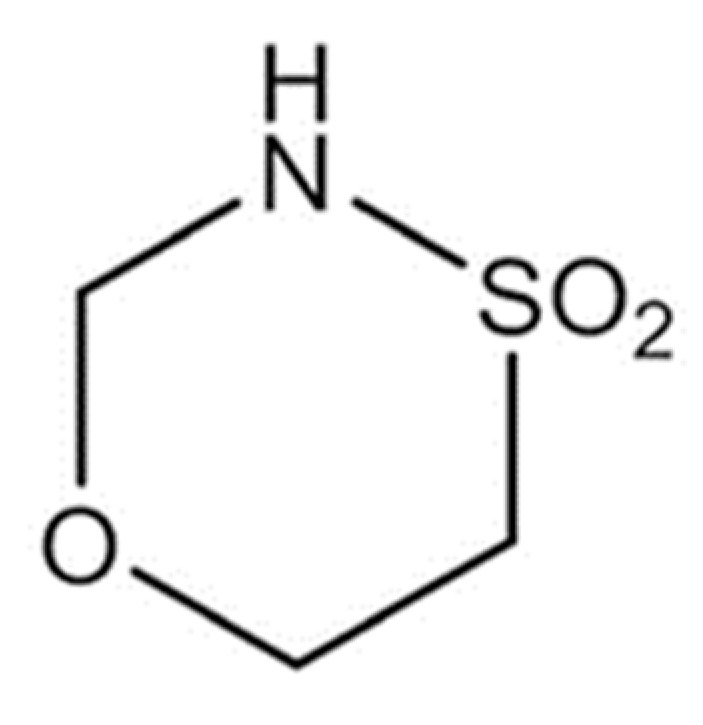
Structural formula of the novel anti-neoplastic agent GP-2250 (Tetrahydro-1,4,5-Oxathiazin-4,4-dioxid); reprinted with permission from [15].

**Figure 2 ijms-23-07293-f002:**
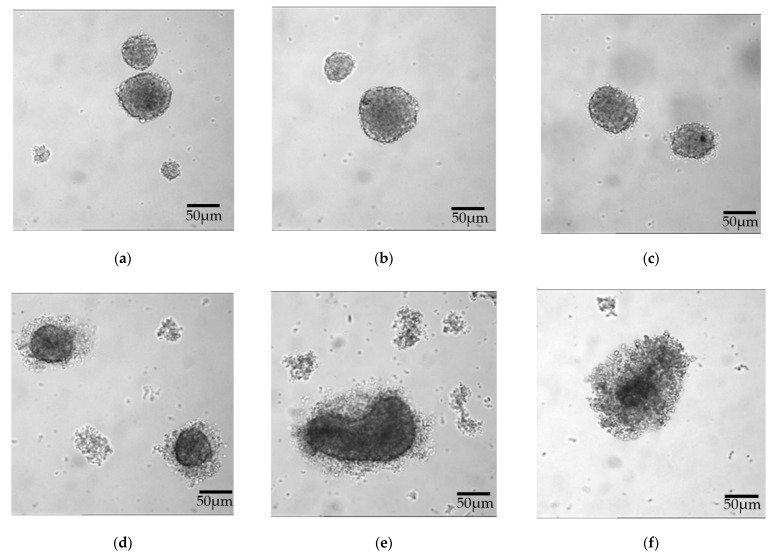
JL-1 sph. after a 48 h treatment with increasing doses of GP-2250: (**a**) control, (**b**) 250 µmol/L GP-2250, (**c**) 500 µmol/L GP-2250, (**d**) 750 µmol/L GP-2250, (**e**) 1000 µmol/L GP-2250 and (**f**) 1500 µmol/L GP-2250. With increasing doses, cell–cell contacts of the spheroid periphery are loosened, and singular cells within the spheroid formations can be differentiated.

**Figure 3 ijms-23-07293-f003:**
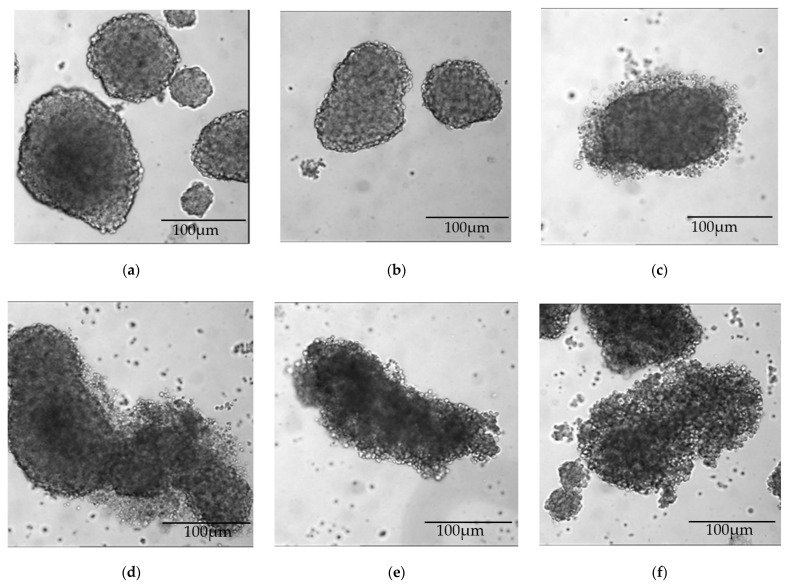
MSTO-211H sph. after 48 h treatment with increasing doses of GP-2250: (**a**) control, (**b**) 500 µmol/L, (**c**) 1000 µmol/L GP-2250, (**d**) 1500 µmol/L GP-2250, (**e**) 2000 µmol/L GP-2250 and (**f**) 2500 µmol/L GP-2250. Similar to JL-1 sph., GP-2250 had a microscopic effect on the spheroid periphery by loosening cell–cell contacts. A dense spheroid core can still be identified wherein single cells cannot be distinguished.

**Figure 4 ijms-23-07293-f004:**
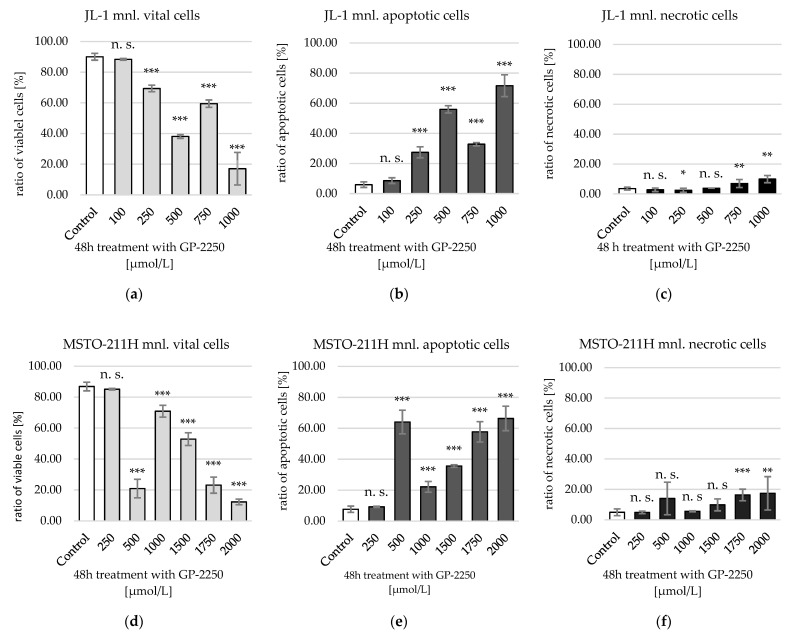
Analysis of cytotoxicity after 48 h treatment of JL-1 and MSTO-211H mnl. cells with increasing doses of GP-2250 and untreated control. Shown are mean values of (**a**) vital, (**b**) apoptotic and (**c**) necrotic JL-1 mnl. cells and (**d**) vital, (**e**) apoptotic and (**f**) necrotic MSTO-211H mnl. cells. Cells were labeled with annexin V-FITC/Propidium Iodide (PI) dual-binding to measure the ratio of vital, apoptotic and necrotic cells via flow cytometry. Values are means ± standard deviation (SD) of at least 3 independent experiments with consecutive passages. Asterisk symbols indicate statistical significance between treatment and control. *** *p* ≤ 0.001, ** *p* ≤ 0.01, * *p* ≤ 0.05, n.s. *p* > 0.05 (unpaired *t*-test).

**Figure 5 ijms-23-07293-f005:**
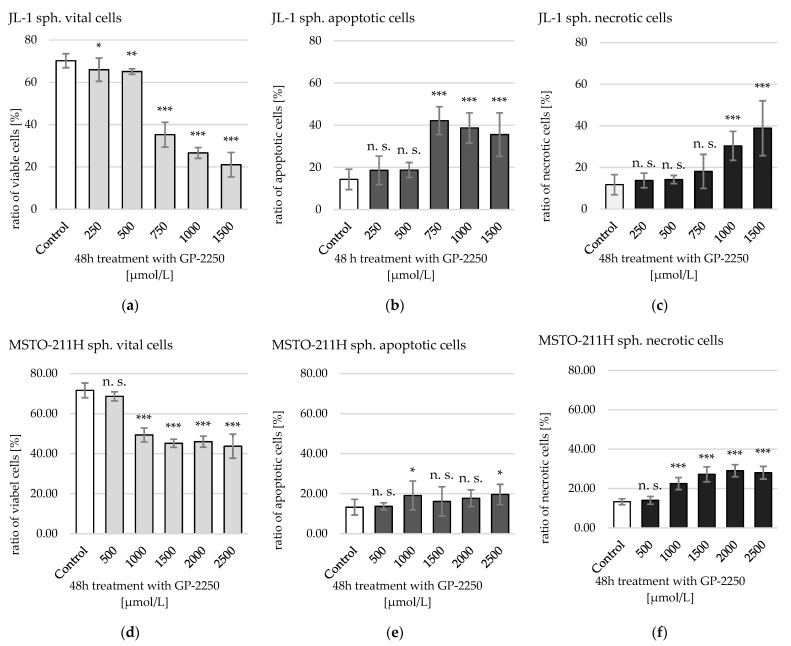
Analysis of cytotoxicity of increasing doses of GP-2250 on JL-1 and MSTO-211H spheroids after 48 h of treatment. Shown are mean values of (**a**) vital, (**b**) apoptotic and (**c**) necrotic JL-1 sph. cells and (**d**) vital, (**e**) apoptotic and (**f**) necrotic MSTO-211H sph. cells. Cells were labeled with annexin V-FITC/ PI dual-binding to measure the ratio of vital, apoptotic and necrotic cells via flow cytometry. Values are means ± SD of at least 3 independent experiments with consecutive passages. Asterisk symbols indicate statistical significance between treatment and control. *** *p* ≤ 0.001, ** *p* ≤ 0.01, * *p* ≤ 0.05, n.s. *p* > 0.05 (unpaired *t*-test).

**Figure 6 ijms-23-07293-f006:**
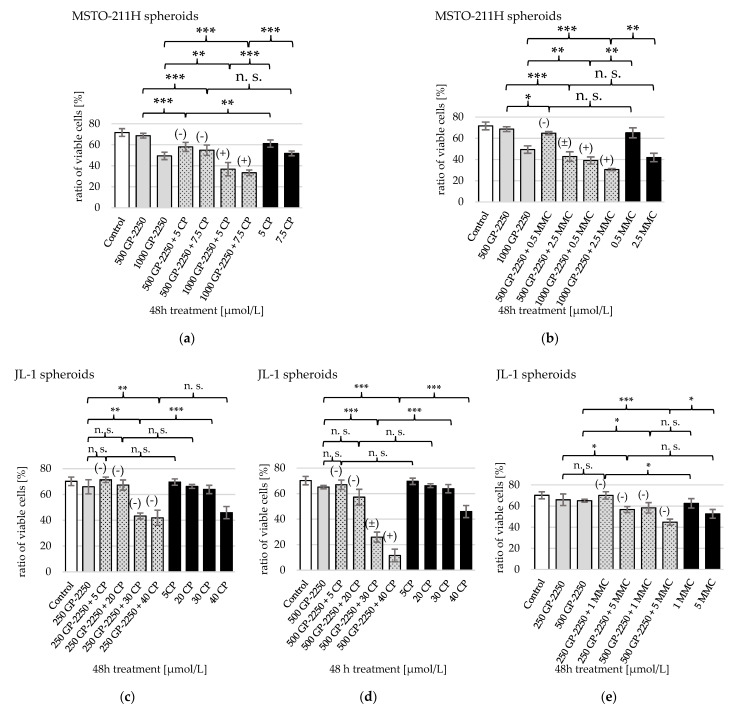
Analysis of cytotoxicity via flow cytometry after 48 h combination treatments. The graphs display the ratio of viable cells of (**a**) MSTO-211H sph. treated with GP-2250 + CP, (**b**) MSTO-211H sph. treated with GP-2250 + MMC, (**c**,**d**) JL-1 sph. treated with GP-2250 + CP, (**e**) JL-1 sph. treated with GP-2250 + MMC. Cells were marked with annexin V-FITC/ PI dual-binding, following the measurement of viable, apoptotic and necrotic cells via flow cytometry. Values are means ± SD of at least 3 independent experiments with consecutive passages. Asterisk symbols indicate the statistical significance between the combination-treatment and their respective monotherapies. *** *p* ≤ 0.001, ** *p* ≤ 0.01, * *p* ≤ 0.05, n.s. *p* > 0.05 (unpaired t-test). The combination indices (CI) by Chou-Talalay are marked as followed: CI < 0,9: symbiotic effect (+), CI = 0.9–1.1: additive effect (±) and CI > 1, 1: antagonistic effect (−).

**Figure 7 ijms-23-07293-f007:**
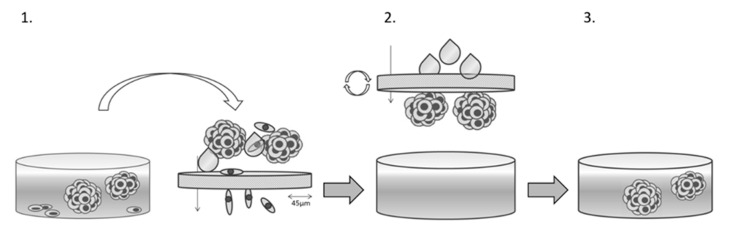
Spheroid formation assay. Cells are cultured in stem cell medium on ultra-low attachment plates. After their respective incubation periods, cultures are filtered over a 45 µmol/L cell filter. Single cells pass through the filter. Cells which are part of a very dense cell cluster that cannot be separated through mechanical means are called spheroids. Spheroids are held back in the retentate. The retentate is then resuspended in stem cell medium and is ready for treatment.

**Table 1 ijms-23-07293-t001:** Doses needed to achieve a decline of viable cells by 50% (ED_50_) or more after a 48 h treatment period.

Culture	Treatment	≥ED_50_ [µmol/L]	
Monolayer		JL-1	MSTO-211H
	GP-2250	603.7 ^1^	1193.04 ^1^
Spheroids	GP-2250	949.64 ^1^	2587.99 ^1^
	MMC	10	5–10
	CP	50	20
	GP-2250 + MMC	x	1000 GP-2250 + 0.5–2.5 MMC
	GP-2250 + CP	500 GP-2250 + 30 CP	1000 GP-2250 + 5–7.5 CP

^1^ Computed using CalcuSyn 2.11.

## Data Availability

The datasets used and analyzed during the current study are available from the corresponding author on reasonable request.

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
