# Peer review of "Substance GP-2250 as a New Therapeutic Agent for Malignant Peritoneal Mesothelioma—A 3-D In Vitro Study"

_ijms, 2022, doi:10.3390/ijms23137293_

Round 1

Reviewer 1 Report

The work of Baron and co-workers describes the cytotoxic effect of a novel anti-cancer therapeutic agent (GP-2250) on two peritoneal mesothelioma cell lines organized as a monolayer and a spheroid culture. In addition, a combination of two established chemotherapeutics (cis-Platin and mitomycin C) was also tested. The results are significant and represent the progress in the field, but prior to its acceptance for publication, several issues need a correction (including additional experiments):

Additional experiments:

-How do the authors explain the higher moiety of necrotic over apoptotic cells in spheroid cells compared to monolayer cell culture? What is the % of necrotic and apoptotic cells in a combined (GP-2250/CP or GP-2250 MMC) therapy? These results should be added. Along with the increased cytotoxicity, these data will complete the results section, strengthen the discussion and make the conclusion more convincing.

-The authors discuss the synergistic effect of ROS generated by GP-2250 and CP, but there is no experimental evidence to support these statements. These results would significantly improve the quality of the manuscript.

Changes:

-Figs 2 and 3-scale bar is missing. Therefore it is difficult to monitor the results section.

-The authors should explain the difference between these two cell lines. Why are these cell lines used for studies?

-The references and a discussion explaining a higher ED50 for both spheroid cell lines are necessary compared to the monolayer cell organization.  

Also, the combination of GP-2250 and CP or MMC at the lowest concentrations does not significantly affect the cytotoxicity of MM spheroids. What is a possible explanation for this (please refer to Figure 6)

-In the legend to Fig. 6. Delete “analysis” (Analysis of cytotoxicity by flow cytometry analysis).

-Legend to Figure 5. “Analysis of cytotoxicity after 48 h treatment of JL-1 and MSTO-211H sph. cells with 156 increasing doses of GP-2250 and untreated control.” It seems not to be used correctly. The cytotoxic effect of indicated drugs is investigated. It would be better to state, “Analysis of cytotoxicity of 156 increasing doses of GP-2250 on JL-1 and MSTO-211H spheroids after 48h of treatment”.

-An increase in the moiety of necrotic vs. apoptotic cells in the spheroids should be addressed. As necrosis as a type of cell death is not desirable, the prevalence of this type of cell death is not the advantage of the therapy. Please, comment in more detail. However, since spheroids are more physiologically relevant and near an experimental clinic setup, the authors must address this issue.

How are the relevant CP and MMC concentrations used to combine with GP-2250 determined? The concentrations of MMC used for the experiments are much lower than concentrations of CP, and a combination of GP-2250 and CP achieves results in a higher ratio of cytotoxic cells. Based on what, these concentrations are selected for Analysis? If separate experiments are performed to select the working concentrations, authors should present them either in the manuscript text or in the supplementary material.

Finally, the Discussion section must be shorter and more focused. 

Author Response

We thank the reviewer for their careful evaluation and insightful comments. We are grateful for giving us the chance to revise the manuscript. Please find below the reviewer’s comments (blue) and our according answers (black).

-How do the authors explain the higher moiety of necrotic over apoptotic cells in spheroid cells compared to monolayer cell culture? What is the % of necrotic and apoptotic cells in a combined (GP-2250/CP or GP-2250 MMC) therapy? These results should be added. Along with the increased cytotoxicity, these data will complete the results section, strengthen the discussion and make the conclusion more convincing.

We thank the reviewer for making this valid point.

As reviewed by Ryu and colleagues, spheroids have several advantages over the monolayer culture and thus allowing further approximation to the in vivo culture. However, a diffusion gradient leads to lack of nutrients in the spheroids’ “core” [1] and thus causing higher shares of necrotic than apoptotic cells in comparison to the monolayer model [2].

We added the shares of necrotic and apoptotic cells under treatment into the result and discussion section and thank the reviewer for this valuable input.

-The authors discuss the synergistic effect of ROS generated by GP-2250 and CP, but there is no experimental evidence to support these statements. These results would significantly improve the quality of the manuscript.

The Reviewer is correct, to date there is no data concerning ROS generation of combination therapies. This is currently subject of an ongoing investigation.

Changes:

-Figs 2 and 3-scale bar is missing. Therefore it is difficult to monitor the results section.

We thank the reviewer for this comment, a scale bar has been added to the figures.

-The authors should explain the difference between these two cell lines. Why are these cell lines used for studies?

We thank the reviewer for pointing this out. For this study we investigated three established and commercially available cell lines. NCI-H28, a human epitheloid mesothelioma cell line established from pleural effusions did not form spheroids and was consequently excluded from analysis. JL-1 a human epitheloid malignant mesothelioma cell line was derived from a 54-year-old male diagnosed with asbestos associated epitheloid pleural mesothelioma and did form spheroids. The other analysed cell line was MST-211, derived from pleura effusions of a 62-year-old male diagnosed with malignant biphasic mesothelioma. A corresponding paragraph has been added to the materials and methods section.

-The references and a discussion explaining a higher ED50 for both spheroid cell lines are necessary compared to the monolayer cell organization.  

We thank the reviewer for making this valid point. As described previously, due to their 3D structure spheroids are subject to a diffusion gradient from the exposed site to the “core” [1], thus higher concentrations of chemotherapeutics are needed in order to expose especially the inner zones to an effective dose. Therefore, the ED50 is higher in spheroids compared to monolayer where all cells are equally exposed to chemotherapy.

Also, the combination of GP-2250 and CP or MMC at the lowest concentrations does not significantly affect the cytotoxicity of MM spheroids. What is a possible explanation for this (please refer to Figure 6)

We thank the author for pointing this out. Considering that combination therapy aims to reduce its single agents’ concentration and thereby adverse effects, effects of very small concentrations have been investigated. Evidently, the combination of the lowest concentrations of CP/MMC with GP-2250 proved to be ineffectively low.

-In the legend to Fig. 6. Delete “analysis” (Analysis of cytotoxicity by flow cytometry analysis).

We appreciate the reviewers’ comment and changed the title to “Analysis of cytotoxicity via flow cytometry after 48h combination-treatments”.

-Legend to Figure 5. “Analysis of cytotoxicity after 48 h treatment of JL-1 and MSTO-211H sph. cells with 156 increasing doses of GP-2250 and untreated control.” It seems not to be used correctly. The cytotoxic effect of indicated drugs is investigated. It would be better to state, “Analysis of cytotoxicity of 156 increasing doses of GP-2250 on JL-1 and MSTO-211H spheroids after 48h of treatment”.

We thank the reviewer for pointing this out and altered the title accordingly.

-An increase in the moiety of necrotic vs. apoptotic cells in the spheroids should be addressed. As necrosis as a type of cell death is not desirable, the prevalence of this type of cell death is not the advantage of the therapy. Please, comment in more detail. However, since spheroids are more physiologically relevant and near an experimental clinic setup, the authors must address this issue.

We thank the reviewer for this valuable input. Treatment of spheroids with a combination therapy of GP 2250 and CP/MMC resulted in an increase of the moiety of both, necrotic and apoptotic cells. As described above, reasons are multi-faceted.

Firstly, similar to monolayer cells, physiological apoptosis is a highly controlled, adenosine triphosphate (ATP)-dependent process with subsequent phagocytosis of the degradation products. In absence of inflammatory cells and thus phagocytosis, in in vitro cell culture secondary necrosis occurs. The presently utilized flow cytometry markers Annexin-V and Propidium Iodide (PI) cannot differentiate between primary and secondary necrosis as PI intercalates with exposed DNA, found in both necrotic and late apoptotic cells. This might lead to false positive measurement of necrotic cells.

Secondly, Buchholz, Majchrzak-Stiller et al. suggest that GP-2250 might initiate programmed necrosis. This process is a caspases-independent programmed cell death initiated by cell stress or activation of death receptors, leading to excessive levels of reactive oxygen species (ROS) [3,4]. GP-2250, in turn, has been shown to increase intracellular ROS levels [3].

Thirdly, as already described above, in spheroids a diffusion gradient of nutrients results in necrosis in the spheroids’ centres [1,2] which is augmented through application of chemotherapeutics.

How are the relevant CP and MMC concentrations used to combine with GP-2250 determined? The concentrations of MMC used for the experiments are much lower than concentrations of CP, and a combination of GP-2250 and CP achieves results in a higher ratio of cytotoxic cells. Based on what, these concentrations are selected for Analysis? If separate experiments are performed to select the working concentrations, authors should present them either in the manuscript text or in the supplementary material.

We thank the reviewer for pointing this out.

The relevant concentrations for CP and MMC to combine with GP-2250 were determined based on previous studies by [5,6]. In each case concentrations were chosen that had a low to moderate effect in monotherapy in order to achieve synergism using the lowest effective concentrations.

The concentrations of MMC used for the experiments are indeed much lower than the concentrations of CP. This can be explained due to the structural differences as well as diverging modes of action of both chemotherapeutics. In HIPEC regimens, for example concentrations of 90mg/m² CP are used as opposed to concentrations of 15mg/m² MMC [7].

Finally, the Discussion section must be shorter and more focused. 

We thank the reviewer for this input, the discussion section has been reworked and correspondingly amended.

References

  1. Ryu, N.-E.; Lee, S.-H.; Park, H. Spheroid Culture System Methods and Applications for Mesenchymal Stem Cells. Cells 2019, 8, 1620, doi:10.3390/cells8121620.
  2. Barisam, M.; Saidi, M.; Kashaninejad, N.; Nguyen, N.-T. Prediction of Necrotic Core and Hypoxic Zone of Multicellular Spheroids in a Microbioreactor with a U-Shaped Barrier. Micromachines 2018, 9, 94, doi:10.3390/mi9030094.
  3. Buchholz, M.; Majchrzak-Stiller, B.; Hahn, S.; Vangala, D.; Pfirrmann, R.W.; Uhl, W.; Braumann, C.; Chromik, A.M. Innovative substance 2250 as a highly promising anti-neoplastic agent in malignant pancreatic carcinoma - in vitro and in vivo. BMC Cancer 2017, 17, doi:10.1186/s12885-017-3204-x.
  4. Buchholz, M. Vergleichende anti-neoplastische Charakterisierung der Substanz 2250 mit ihrer Muttersubstanz Taurolidin: - in vitro und in vivo. Dissertation; Ruhr- Universität Bochum, Bochum, 2016.
  5. Zhou, Q.-M.; Wang, X.-F.; Liu, X.-J.; Zhang, H.; Lu, Y.-Y.; Huang, S.; Su, S.-B. Curcumin improves MMC-based chemotherapy by simultaneously sensitising cancer cells to MMC and reducing MMC-associated side-effects. Eur. J. Cancer 2011, 47, 2240–2247, doi:10.1016/j.ejca.2011.04.032.
  6. Cortes-Dericks, L.; Froment, L.; Boesch, R.; Schmid, R.A.; Karoubi, G. Cisplatin-resistant cells in malignant pleural mesothelioma cell lines show ALDHhighCD44+ phenotype and sphere-forming capacity. BMC Cancer 2014, 14, doi:10.1186/1471-2407-14-304.
  7. Sugarbaker, P.H.; van der Speeten, K. Surgical technology and pharmacology of hyperthermic perioperative chemotherapy. J. Gastrointest. Oncol. 2016, 7, 29–44, doi:10.3978/j.issn.2078-6891.2015.105.

Reviewer 2 Report

The ms by Claudia Baron and co-workers explores the effects ofGP2250 alone or in combination on malignant mesothelioma. The authors indicate on line 21 peritoneal mesothelioma, but MSTO211H cells are from pleural effusions, and JL1 cell line is a pleural epithelioid mesothelioma. The authors utilized only two mesothelioma cell lines. No comparison on toxicity with a normal mesothelial cells is performed. 

Only citotoxicity data on spheroid and monolayers are available, also for combination treatment with mytomicin and cisplatin.

No idea about molecular mechanisms or synergistic behaviour are available.

Discussion is too long. Statistical analysis should better defined.

Author Response

We thank the reviewer for their thorough assessment. We are grateful for giving us the chance to revise the manuscript. Please find below the reviewer’s comments (blue) and our according answers (black).

The authors indicate on line 21 peritoneal mesothelioma, but MSTO211H cells are from pleural effusions, and JL1 cell line is a pleural epithelioid mesothelioma. The authors utilized only two mesothelioma cell lines. No comparison on toxicity with a normal mesothelial cells is performed. 

We thank the reviewer for pointing this out, evidently there has been a mistake in the abstract, to avoid further confusion we altered the sentence accordingly. Originally, analysis included three mesothelioma cell lines, NCI-H28, JL-1 and MSTO-211. However, NCI-H28 did not form spheroids and were therefore excluded.

As GP-2250 proved to be well tolerated in mice and in a recent Phase I/II study [1], a control on non-pathological mesothelial cells was not performed. As shown in previous studies of our working group, even direct intraperitoneal application in mice did not result in impairment of mesothelial function or morphological changes of the peritoneum [2,3].

Only citotoxicity data on spheroid and monolayers are available, also for combination treatment with mytomicin and cisplatin.

We thank the reviewer for pointing this out, this is the first study to investigate the effects of GP-2250 alone and in combination with CP and MMC. In vivo data is needed, however ethical approval depends on published data.

No idea about molecular mechanisms or synergistic behaviour are available.

We appreciate the reviewer’s statement, the substance GP-2250 is a very recent development, data on its mechanism of action is pending.

Discussion is too long. Statistical analysis should better defined.

We thank the reviewer for this comment, the discussion has been completely revised and statistical analysis has been clearer defined.

References

  1. Kasi, A.; Iglesias, J.L. A phase 1/2 study to evaluate the safety, tolerability, and preliminary efficacy of GP-2250 in combination with gemcitabine for advanced or metastatic pancreatic adenocarcinoma. J. Clin. Oncol. 2022, 40, TPS620-TPS620, doi:10.1200/JCO.2022.40.4_suppl.TPS620.
  2. Buchholz, M.; Majchrzak-Stiller, B.; Hahn, S.; Vangala, D.; Pfirrmann, R.W.; Uhl, W.; Braumann, C.; Chromik, A.M. Innovative substance 2250 as a highly promising anti-neoplastic agent in malignant pancreatic carcinoma - in vitro and in vivo. BMC Cancer 2017, 17, doi:10.1186/s12885-017-3204-x.
  3. Buchholz, M. Vergleichende anti-neoplastische Charakterisierung der Substanz 2250 mit ihrer Muttersubstanz Taurolidin: - in vitro und in vivo. Dissertation; Ruhr- Universität Bochum, Bochum, 2016.

Round 2

Reviewer 1 Report

The authors have improved the paper, but some issues still need to be addressed before final acceptance. 

  1. The role of ROS in the process of apoptosis is an important issue. Although the authors state that the evaluation of the ROS production is an ongoing investigation, I still think these results will support the data and introduce the mechanism of action. If authors have any preliminary data that might support this statement, they are free to enclose them. 
  2. The reply to my comment, “-The references and a discussion explaining a higher ED50 for both spheroid cell lines are necessary compared to the monolayer cell organization,” is not finished. Please, address this issue in the manuscript text, i.e., in the discussion section. 
  3. The authors reply: “The relevant concentrations for CP and MMC to combine with GP-2250 were determined based on previous studies [5,6]. In each case, concentrations that had a low to moderate effect in monotherapy were chosen to achieve synergism using the lowest effective concentrations.” Moreover, the explanation of the lower concentrations of MMC used in experiments should be involved in the manuscript text as an explanation and for clarity reasons. . If the authors have already done that, I apologize for not being able to find this paragraph. 
  4. The authors must be more careful in citing new references. (e.g., second paragraph in the Discussion section). 

Author Response

Manuscript ID IJMS- 1766745

Response to Reviewer 1

Dear Reviewer 1,

We thank you for dedicating your time and effort for this careful re-assessment of our work. We are grateful for the insightful comments on and valuable improvements to our paper.

We have incorporated the suggestions made. Those changes are highlighted within the manuscript.

Please see below for a point-by-point response to your comments and concerns.

The role of ROS in the process of apoptosis is an important issue. Although the authors state that the evaluation of the ROS production is an ongoing investigation, I still think these results will support the data and introduce the mechanism of action. If authors have any preliminary data that might support this statement, they are free to enclose them. 

Response to 1.:

We thank you for this comment. We have preliminary data for the pancreatic carcinoma cell line PancTuI. They indicate a synergistc effect of GP-2250 and Cisplatin on ROS response. If requested we are able to generate the according data at least in two mesothelioma cell lines within two weeks.

The reply to my comment, “-The references and a discussion explaining a higher ED50 for both spheroid cell lines are necessary compared to the monolayer cell organization,” is not finished. Please, address this issue in the manuscript text, i.e., in the discussion section. 

Response to 2.:

We thank you for this valid point, the corresponding paragraph in the discussion was revised and elaborated, see line 346-356 in the manuscript.

“Concentrations needed to achieve an ED50 were higher in spheroids than in monolayers of both cell lines. While in monolayer all cells are equally exposed to media and chemotherapeutics, this is not the case in 3-D structured spheroids. Thus, higher concentrations of chemotherapeutics are needed in order to expose especially the inner zones to an effective dose. Furthermore, the increased cellular adhesion of tumor cells in spheroids has been proven to reduce sensitivity towards cytotoxic drugs [1–4]. “

The authors reply: “The relevant concentrations for CP and MMC to combine with GP-2250 were determined based on previous studies [5,6]. In each case, concentrations that had a low to moderate effect in monotherapy were chosen to achieve synergism using the lowest effective concentrations.” Moreover, the explanation of the lower concentrations of MMC used in experiments should be involved in the manuscript text as an explanation and for clarity reasons. If the authors have already done that, I apologize for not being able to find this paragraph. 

Response to 3.:

You are right and stating a valid point, a corresponding paragraph has been added into the result sections 2.3. (line 177-180) and 2.4. (line 227-230) of the combination therapies.

The authors must be more careful in citing new references. (e.g., second paragraph in the Discussion section). 

Response to 4.:

You are absolutely correct. The references have been thoroughly revised and corrected; we apologize for the lack of carefulness.

References

  1. Bates, R. Spheroids and cell survival. Critical Reviews in Oncology/Hematology 2000, 36, 61–74, doi:10.1016/S1040-8428(00)00077-9.
  2. St Croix, B.; Flørenes, V.A.; Rak, J.W.; Flanagan, M.; Bhattacharya, N.; Slingerland, J.M.; Kerbel, R.S. Impact of the cyclin-dependent kinase inhibitor p27Kip1 on resistance of tumor cells to anticancer agents. Nat. Med. 1996, 2, 1204–1210, doi:10.1038/nm1196-1204.
  3. Graham, C.H.; Kobayashi, H.; Stankiewicz, K.S.; Man, S.; Kapitain, S.J.; Kerbel, R.S. Rapid acquisition of multicellular drug resistance after a single exposure of mammary tumor cells to antitumor alkylating agents. J. Natl. Cancer Inst. 1994, 86, 975–982, doi:10.1093/jnci/86.13.975.
  4. Kobayashi, H.; Man, S.; Graham, C.H.; Kapitain, S.J.; Teicher, B.A.; Kerbel, R.S. Acquired multicellular-mediated resistance to alkylating agents in cancer. Proc. Natl. Acad. Sci. U. S. A. 1993, 90, 3294–3298, doi:10.1073/pnas.90.8.3294.
  5. Zhou, Q.-M.; Wang, X.-F.; Liu, X.-J.; Zhang, H.; Lu, Y.-Y.; Huang, S.; Su, S.-B. Curcumin improves MMC-based chemotherapy by simultaneously sensitising cancer cells to MMC and reducing MMC-associated side-effects. Eur. J. Cancer 2011, 47, 2240–2247, doi:10.1016/j.ejca.2011.04.032.
  6. Cortes-Dericks, L.; Froment, L.; Boesch, R.; Schmid, R.A.; Karoubi, G. Cisplatin-resistant cells in malignant pleural mesothelioma cell lines show ALDHhighCD44+ phenotype and sphere-forming capacity. BMC Cancer 2014, 14, doi:10.1186/1471-2407-14-304.

Reviewer 2 Report

The authors have correctly replied to comments

Author Response

Manuscript ID IJMS- 1766745

Response to Reviewer 2

Dear Reviewer 2,

We thank you for dedicating your time and effort for this careful re-assessment of our work. We are grateful for the insightful comments on and valuable improvements to our paper.

We have incorporated the suggestions made. Those changes are highlighted within the manuscript.

Please see below for a poini-by-point response to your comments and concerns.

Below you will find the reviewer’s comments and our corresponding responses.

Reviewer:

The authors have correctly replied to comments.

Response:

Thank you very much for your comment and helping us to improve the manuscript.
